



# From column to surface: connecting the performance in simulating aerosol optical properties and PM2.5 concentrations in the NASA GEOSCCM

Caterina Mogno[1,2], Peter R. Colarco[2], Allison B. Collow[1,3], Sampa Das[2,4], Sarah A. Strode[2,5], Vanessa Valenti[6,7], Michael E. Manyin[2,6], Qing Liang[2], Luke Oman[2], Stephen D. Steenrod[1,2], K. Emma Knowland[3,5,8]

[1] Goddard Earth Sciences Technology and Research II (GESTAR II), University of Maryland Baltimore County, Baltimore, Maryland, USA

[2] Atmospheric Chemistry and Dynamics Lab, NASA Goddard Space Flight Center, Greenbelt, Maryland, USA

[3] Global Modeling and Assimilation Office, NASA Goddard Space Flight Center, Greenbelt, Maryland, USA

[4] Earth System Science Interdisciplinary Center (ESSIC), University of Maryland, College Park, Maryland, USA

[5] Goddard Earth Sciences Technology and Research II (GESTAR II), Morgan State University, Baltimore, Maryland, USA

[6] Science Systems and Applications, Inc., Lanham, Maryland, USA

[7] Computational and Information Sciences and Technology Office, NASA Goddard Space Flight Center Greenbelt, Maryland, USA

[8] now at: NASA Headquarters, Washington, District of Columbia, USA

*Correspondence to*: Caterina Mogno (caterina.mogno@nasa.gov)

**Abstract.** Aerosols are a key climate forcer and harmful to human health at the surface. Accurately modeling aerosol optical properties, mass loading and their relationship is important for constraining aerosol-climate forcing and characterizing particulate matter pollution exposure. We investigate the drivers of uncertainties in the NASA Goddard Earth Observing System Chemistry Climate Model (GEOSCCM) in simulating aerosols by focusing on the link between aerosol optical properties and mass. We compare a GEOSCCM hindcast with long-term coincident observations including satellite AOD measurements, speciated PM2.5 datasets from observations-model data fusion, and ground-based measurements of aerosol mass and optical properties. We analyze regional trends and seasonal variations of AOD and PM2.5, and surface aerosol properties, including relative humidity's role in hygroscopic enhancement. This work also presents the first extensive assessment of GEOSCCM's aerosol component with observational data. Our findings show that biases in PM2.5 components and relative humidity significantly impact simulated aerosol scattering at the surface, while scattering efficiency assumptions align with observations. This indicates that errors in simulated scattering relate more to simulated aerosol mass and relative humidity than optical properties and size distribution assumptions in GEOSCCM. Our work highlights the importance of relative humidity biases on aerosol scattering enhancement for climate models where meteorology is not prescribed. Findings suggest improvements in GEOSCCM aerosols mass and optical properties could be achieved through updating emission inventories, especially over biomass burning regions, reducing nitrate biases, and improving relative humidity simulation.



## 1 Introduction

Aerosols are an important climate forcer and are harmful pollutants to human health. Aerosols impact climate directly through scattering and absorption of solar radiation (aerosol-radiation interaction) and indirectly by acting as nuclei for cloud droplet and ice particle formation (aerosol-cloud interactions) (IPCC 2023). Uncertainties in aerosol properties, distributions, and processes are the main driver of uncertainty in the overall anthropogenic forcing of the climate system, with impacts of aerosol-radiation and aerosol-clouds interactions remaining one of the largest drivers of uncertainty in anthropogenic climate forcing (J. Li et al. 2022; Kahn et al. 2023; IPCC 2023). Close to Earth's surface, exposure to ambient fine particulate matter, i.e. aerosols with diameter less than 2.5 μm ($PM_{2.5}$), is a leading contributor to the global burden of disease. It has been associated with short-term acute health effects and long-term chronic respiratory and cardiovascular diseases, accounting for more than 4 million premature deaths per year (Fuller et al. 2022; Brauer et al. 2024).

The ability of atmospheric models to correctly represent aerosols and their properties is thus fundamental for constraining aerosol climate forcing and projecting the impacts of changing emissions of aerosols and trace gases on the state of the climate, as well as for characterizing the distribution and changes in particulate matter pollution exposure.

The interaction of atmospheric aerosols with radiation is complex and depends on factors such as the aerosol mass, composition, size, shape, mixture, and hygroscopicity (IPCC 2023). Current atmospheric models generally account for the aerosol-radiation interaction by pre-calculating aerosol extinction, scattering and absorption of radiation and their dependence on relative humidity through look-up optical tables (LUT), which are used to convert simulated aerosol mass to optical quantities. Recently, atmospheric models have been utilized in combination with satellite measurements of aerosol optical depth (AOD) to produce satellite-derived estimates of surface $PM_{2.5}$ (Xu et al. 2015; van Donkelaar et al. 2021; 2019; Hammer et al. 2020). The linking of AOD (an optical column measurement) to surface concentrations remains a key uncertainty in these derivations, because it is addressed by using model-based assumptions of the $PM_{2.5}$/AOD ratio, which is subject to further assumptions in the model representation of aerosol processes, properties, vertical distribution and hygroscopic enhancement (Zhai et al. 2021). Inaccuracies in modeled $PM_{2.5}$/AOD ratios can propagate into errors in satellite-derived surface $PM_{2.5}$ estimates, especially in regions with complex aerosol mixtures or vertical distribution (Zhu et al. 2024). In particular, recent aerosol model intercomparison and evaluation efforts highlight the need to investigate model biases by looking beyond standard aerosol global optical depth (AOD) evaluations and incorporating additional analysis such as the characterization of regional and seasonal variations, aerosol hygroscopic growth, and whether model biases in optical depth are linked to biases in aerosol mass or optical property assumptions (Gliß et al. 2021).

In this context, the use of simultaneous observations from various sources are essential to improve our representation of modeled aerosol-radiation interactions and of particulate matter. The expanding atmosphere observing system, which includes satellite, aircraft and ground-based platforms, can provide a wide range of long-term, complementary and coincident aerosol measurements. The simultaneous use of these observations can enhance our understanding and interpretation of observed and modeled aerosol burdens, the relationships between their optical, physical, and chemical properties (Mortier et al. 2020;



Bharath et al. 2024; Kahn et al. 2023). Specifically, co-located aerosol chemistry and size data, relative humidity, and light scattering measurements can be used to better constrain model aerosol optical property assumptions (Latimer and Martin 2019; Burgos et al. 2020).

In this work we investigate the drivers of uncertainties in the current NASA Goddard Earth Observing System Chemistry Climate Model (GEOSCCM) in simulating aerosols by focusing on the link between aerosol optical properties and mass. We accomplish this by comparing a GEOSCCM hindcast with a variety of long-term coincident observations of aerosol, including satellite measurements of AOD, datasets of speciated $PM_{2.5}$ from observation-model data fusion products, and ground-based measurements of aerosol mass and optical properties. We focus our analysis on regional trends and seasonal variations of AOD and $PM_{2.5}$, and on the properties of aerosols at the surface. In particular, we examine speciated $PM_{2.5}$ and the consistency of GEOSCCM $PM_{2.5}$ simulation with optical properties, including the role of relative humidity in hygroscopic scattering enhancement, using surface observations of co-located long-term speciated $PM_{2.5}$, scattering coefficient, and relative humidity. This is also the first time that the GEOSCCM aerosol component is extensively benchmarked against observations.

In Section 2 we introduce the GEOSCCM model, the GOCART aerosol module and the simulation analyzed in this study. Section 3 describes the aerosol observation datasets used for evaluating GEOSCCM. In Section 4 we present our results. Finally, Section 5 present the final discussion and conclusions.

## 2 Model Description

### 2.1 GEOSCCM and the GOCART Aerosol Module

GEOSCCM is a configuration of the NASA Goddard Earth Observing System (GEOS) global atmospheric general circulation model (AGCM; Molod et al. 2015) with coupled and radiatively interactive atmospheric chemistry and aerosols. GEOSCCM has been extensively evaluated for stratospheric ozone photochemistry and transport processes in various model intercomparison studies (Morgenstern et al. 2017; Eyring T. 2010), and has been used in a variety of studies to investigate trends in stratospheric chemistry and dynamics (F. Li et al. 2018; L. D. Oman et al. 2010; L. Oman et al. 2008), tropospheric ozone (Liu et al. 2022; Strode et al. 2019; 2017), and impacts of aerosols on atmospheric chemistry and climate (Case et al. 2024; Rollins et al. 2017; Aquila et al. 2014). For this study we use GEOSCCM version Icarus-3_2_MEM_22x.

The GEOS AGCM uses a finite volume dynamical core on a cubed sphere based on (Putman and Lin 2007). The model has 72 vertical layers, with hybrid-sigma coordinates, transitioning from terrain-following at the surface to pressure levels at 180 hPa and a model top at ~80 km (0.01 hPa). The convective scheme is based on a modified version of the relaxed Arakawa-Schubert parametrization (Moorthi and Suarez 1992), while the turbulence parameterization is based on (Lock et al. 2000; Louis 1979). Radiative processes are described by (Chou, M. and Suarez, M. J. 2001) for longwave radiation and (Chou, M.



and Suarez, M. J. 1999) for shortwave radiation. Fluxes at the land/atmosphere interface are determined using the Catchment Land Surface Model (Koster et al. 2000), while the surface layer is determined as in (Helfand and Schubert 1995).

The current GEOSCCM model couples the Global Modeling Initiative (GMI) chemistry mechanism (Nielsen et al. 2017; Strahan, Duncan, and Hoor 2007; Duncan et al. 2007; Douglass et al. 2004) with the Goddard Chemistry, Aerosol, Radiation, and Transport (GOCART) aerosol module (M. Chin et al. 2014; P. Colarco et al. 2010; Mian Chin et al. 2002). The GMI module includes 120 species and over 400 reactions, combining tropospheric and stratospheric chemistry.

     GOCART treats externally mixed dust, sea salt, sulfate, nitrate, and black and organic carbon aerosol species. Dust and
sea salt are each partitioned into five non-interacting size bins and have surface windspeed-dependent source functions. Black and organic carbon aerosols are partitioned into hydrophobic and hydrophilic modes, with anthropogenic and biomass burning sources for both and a biogenic source for organic aerosol. A constant ratio of 1.8 is assumed between organic mass and organic carbon (OM:OC). GOCART utilizes a reduced-complexity secondary organic aerosol (SOA) mechanism that relates volatile organic carbon (VOC) emissions to CO emissions across anthropogenic, biofuel and biomass burning sectors, as established
by (Kim et al. 2015). The generated SOA is allocated to hydrophilic components of organic carbon. Additionally, biogenic sources of SOA are from an online version of the Model of Emissions of Gases and Aerosols from Nature (MEGAN). Throughout the rest of the paper, we refer to model OA as the organic aerosol contributions from all primary and secondary sources (biomass, biogenic, biofuel, and anthropogenic). The sulfur mechanism in GOCART includes tracers for dimethyl sulfide (DMS), sulfur dioxide ($SO_2$), sulfate aerosol ($SO_4$), and methanesulfonic acid (MSA). DMS has surface wind-speed
dependent emissions based on sea water DMS concentrations. $SO_2$ has direct emissions from biomass burning, volcanic degassing and eruptions, and anthropogenic sources from aircraft, ships, and land-based pollution sources. Sulfate has direct emissions from ships but is otherwise a product of oxidation of precursor DMS and $SO_2$ species following the mechanism in (Mian Chin et al. 1996). The nitrate mechanism is described in Huisheng Bian et al. 2017 and tracks ammonia ($NH_3$), bulk ammonium aerosol, and three sizes of nitrate aerosol. There are anthropogenic and biomass burning sources of ammonia.
Ammonium and fine-mode nitrate production are calculated in an equilibrium thermodynamic module that balances water, sulfur, and nitrogen species. Coarse mode nitrate is produced from heterogeneous reaction of nitric acid ($HNO_3$) on sea salt and dust surfaces. GOCART aerosol optical properties are pre-computed for each aerosol tracer and saved in look up tables (LUTs). Optical properties are primarily based on Mie calculations with most parameters as in Mian Chin et al. 2002. Non-spherical dust optical properties are as in P. R. Colarco et al. 2014. In the version of GEOSCCM evaluated here, GOCART
aerosols provide surfaces for heterogeneous chemistry and impact photolysis rates in GMI. GMI oxidants and nitric acid are one-way coupled to GOCART sulfate and nitrate algorithms.

**2.2 The Ref-D1 Experiment**

     The simulation analyzed in this study is a global hindcast run at performed with c90 (~1°) horizontal resolution covering the period 1960 – 2018. The simulation follows the protocol for the Chemistry Climate Model Initiative





(https://blogs.reading.ac.uk/ccmi/) "Ref-D1" experiment (IGAC/APARC, Ref-D1 2022) and was submitted as input to the 2022 WMO Scientific Assessment of Ozone Depletion (WMO 2022). The simulation is performed with a free running atmosphere and prescribed sea surface temperature (SST) and sea ice concentrations (SICs) from the global HadISST1 data set as monthly mean boundary conditions (Rayner et al. 2003). The Ref-D1 simulation includes anthropogenic emissions provided for the Coupled Modelling Intercomparison project Phase 6 (CMIP6), which are from the Community Emissions

Data System (Hoesly et al. 2018) for years 1960-2014, and from the CMIP6 ScenarioMIP experiment (SSP2-4.5 scenario) for the years 2015-2018 (Gidden et al. 2019). Biomass burning emissions for years 1960-2015 are also from CMIP6, which are derived from harmonizing different emissions inventories and models as described in (van Marle et al. 2017). In this dataset satellite-derived biomass-burning emissions from the Global Fire Emissions Database version 4 with small fires (GFED4.1s) are used from 1997 to 2015 (Randerson, J.T et al. 2018). For the remainder years not covered by CMIP6 emissions (2016 to

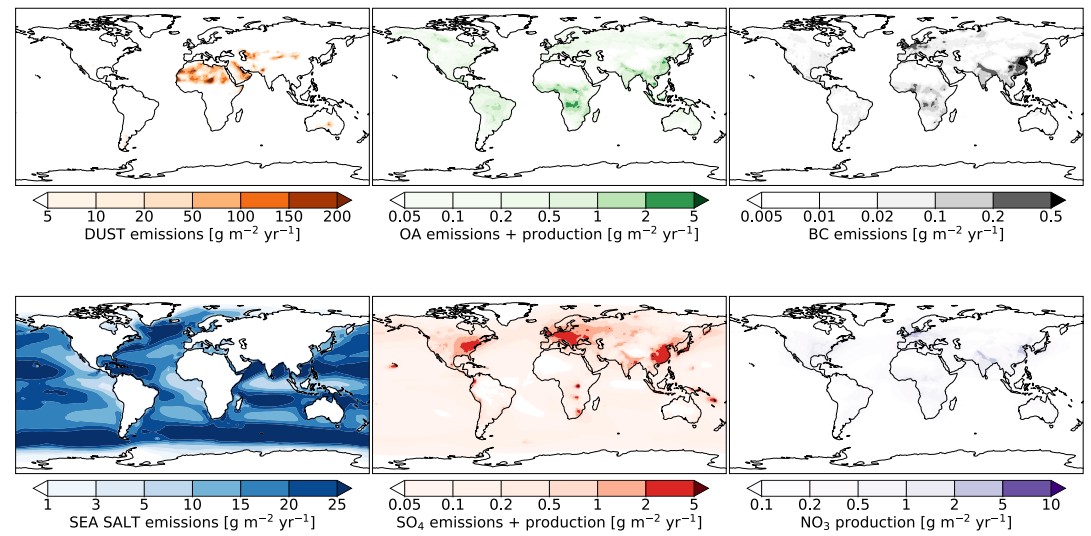


**Figure 1: Global spatial distribution of emissions and chemical production of aerosol species in our Ref-D1 experiment, averaged from 1960-2018. SO₄ includes direct emissions + production, NO₃ refers to production. OA includes all organic mass (emissions + SOA production).**




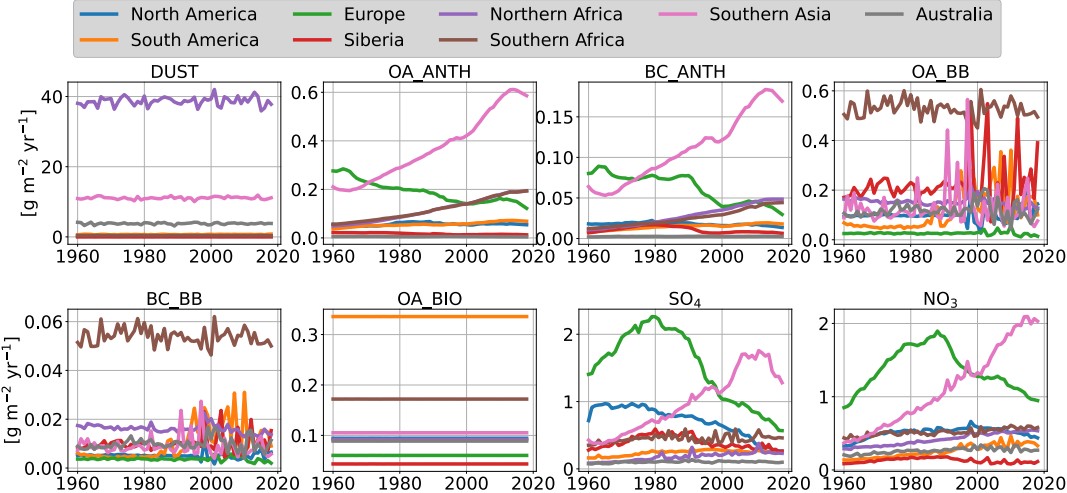

**Figure 2. Ref-D1 regional annual timeseries of emissions and production of aerosols by species for the period 1960-2018. ANTH=anthropogenic, BB=biomass burning, BIO=biogenic. SO₄ includes direct emissions + production but no emissions from volcanic eruptions, NO₃ refers to production. OA includes all organic mass (emissions + production).**

2018), biomass burning emissions are also obtained from GFED4.1. Both anthropogenic and biomass burning emissions include seasonal (monthly) and interannual variability. Finally, for natural emissions sources, biogenic emissions are calculated online in the model using the Model of Emissions of Gases and Aerosols from Nature (MEGAN) (Guenther et al. 2006). Dust and sea salt sources are wind driven following Ginoux et al. 2001 and Gong et al. 2003, Jaeglé et al. 2011, respectively. Volcanic emissions (eruptive + outgassing) follow (Carn et al. 2017). The model also uses prescribed species for aerosol chemistry not calculated by GMI: open-ocean emission of $NH_3$ (Bouwman et al. 1997) and dimethyl sulfide (DMS), whose concentrations are prescribed with seasonally varying climatology and emissions are calculated based on wind speed (Lana et al. 2011; Liss and Merlivat 1986).

Because the Ref-D1 is not driven by real meteorology, we focus our analysis on monthly and yearly timescales. For regional analysis over land, we divide the globe in 8 geographical regions: North America, South America, Northern Africa, Southern Africa, Europe, Siberia, Southern Asia, Australia. The geographical boundaries for the region are reported in Supplemental Figure 1. Regional statistics have been calculated as area-weighted averages of model grid-boxes falling within the boundary for each region. The mathematical definition of the metrics used for evaluating the model against observation, namely normalized mean bias (NMB), root mean square error (RMSE) and Pearson's correlation coefficient (r), are reported in the Supplement S1.





170

175

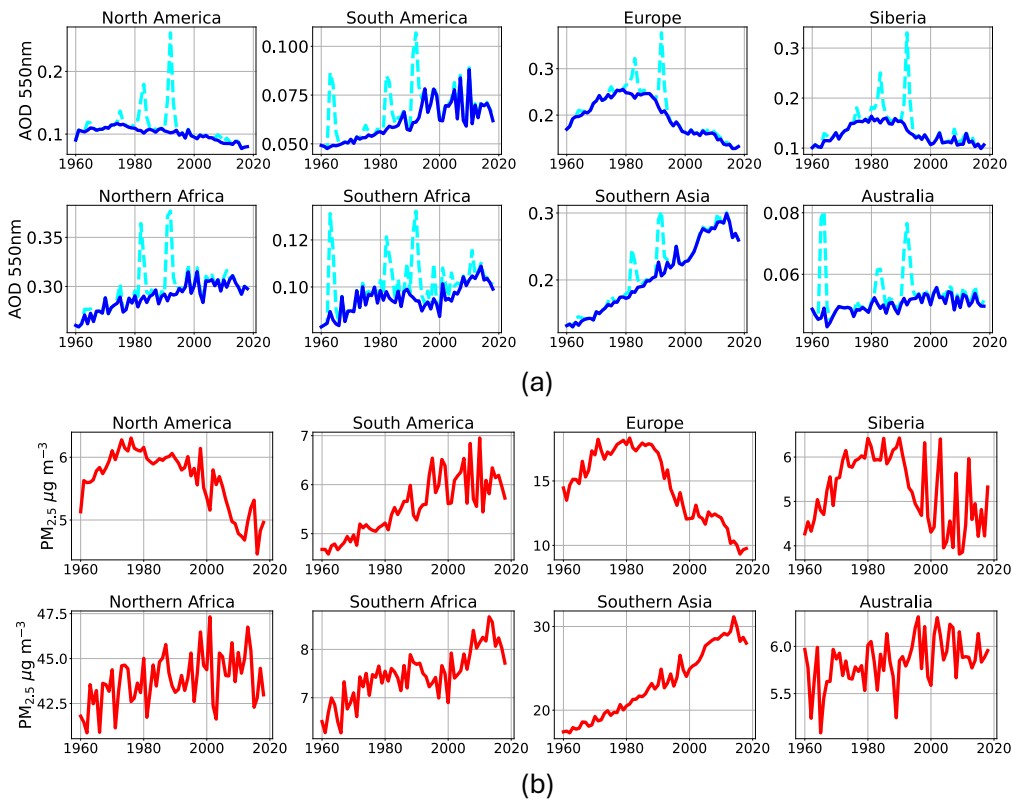

**Figure 3: Ref-D1 regional annual timeseries of (a) total column AOD with (cyan dashed) and without (solid blue) the contribution of volcanic eruptions and (b) surface PM$_{2.5}$ for the period 1960-2018.**



### 2.3 Modelled long-term time-series of emissions, AOD and PM$_{2.5}$

Figure 1 presents the spatial distribution of different aerosol species in Ref-D1, while Figure 2 shows their regional
timeseries. Timeseries of emissions by region are reported in Supplemental Figure 2. Emissions from biomass burning
(OA_BB, BC_BB) shows higher variability and magnitude starting in the late 1990s, reflecting the use of satellite observations
in informing biomass burning estimates from that point forward.

Figure 3 presents regional annual timeseries in Ref-D1 for total AOD and surface PM$_{2.5}$, while Supplemental Figure 3
show the timeseries for speciated AOD and PM$_{2.5}$. Spikes in regional AOD are driven by major volcanic events (e.g. Agung
1963, El Chicon 1982, Pinatubo 1991, Supplemental Figure 3a). Regional changes and trends in emissions are reflected in the
regional AOD and surface PM$_{2.5}$ trends. The decline in anthropogenic SO$_4$ AOD since the 1980s drives a steady decline in
total AOD over Europe, and to a lesser extent over North America, where total AOD has been slowly decreasing. Over Siberia
the decline in SO$_4$ AOD is counterbalanced by the increase of OA AOD from biomass burning since the early 2000s. On the
other hand, increase in SO$_4$ and NO$_3$ AOD drive the increase in total AOD in Southern Asia since the 1960s. Dust AOD is a
relatively constant but important contribution in this region. Total AOD over Southern Asia peaked in the 2010s coincident
with the peak in SO$_4$ and carbonaceous aerosol emissions (Figure 2). Southern Africa and South America also show an increase
in total AOD, driven by increase in OA AOD from biomass burning. In these regions SO$_4$ AOD is also an important steady
contributor to total AOD. Northern Africa and Australia AOD are dominated by natural component of dust and sea salt,
resulting in an almost constant total AOD in the last decades.

Similar to total AOD, the declines in PM$_{2.5}$ in Europe, North America and Siberia starting from the 1980s are driven
mainly by the declines in anthropogenic SO$_4$ PM$_{2.5}$, which was more than halved in all the regions between the 1990s and late
2010s. Further decline in surface PM$_{2.5}$ is driven by reduction in the NO$_3$ and NH$_4$ components in Europe and Siberia, and to
a lesser extent North America, where NH$_4$ exhibit a slow decline while the NO$_3$ component remained stable. These declines
are partially compensated in Siberia and North America by the increase in magnitude and variability of OA from biomass
burning emissions (Figure 2). The OA component in PM$_{2.5}$ is also the dominant component in Southern Africa and South
America, driving the total increasing trends in total PM$_{2.5}$ since the 1960s and the early 2000s respectively. As for AOD, PM$_{2.5}$
in Southern Asia increased mainly due to the increase in SO$_4$, NO$_3$ and OA. This increase is on top of a generally constant
contribution from fine dust which dominates PM$_{2.5}$ in the beginning of the time series.

### 3 Observations and methodology used for model evaluation

Here we describe in detail each dataset and the methodology used in the evaluation of mass and optical properties of
aerosols in the Ref-D1 run. Table 1 summarizes the observations used.



### 3.1 Satellite AOD from MODIS Neural Net Retrieval (NNR)

We compare the global and regional modelled total column AOD with observations of total column AOD at 550 nm retrieved by the Moderate Resolution Imaging Spectroradiometer (MODIS) onboard the Aqua spacecraft. The version of the dataset used for the comparison is based on the Neural Network Retrieval (NNR), a bias-corrected and quality-controlled retrieval of AOD based on MODIS reflectance observations suitable for use in aerosol data assimilation (Randles et al. 2017). The NNR approach retrieves AOD using cloud-screened and homogenized reflectances from the standard Dark Target (Levy et al. 2013) and Deep Blue (Hsu et al. 2013) retrieval algorithms that have been calibrated to co-located AOD measurements from the Aerosol Robotic Network (AERONET, see Section 3.2) using a neural network algorithm. Because our experiment is a free-running model and no data assimilation is invoked we are here comparing the statistical representation of aerosol interannual, seasonal, monthly, and regional variability in our model run to data rather than focusing on specific events.

We obtain NNR data global monthly means by averaging all the overpasses within each month, and regrid it from its native resolution (0.25° x 0.3125°) to the model spatial resolution. For each month, the model outputs have been spatially sampled to match where monthly NNR data were available. This approach is appropriate given that the model is not run with real meteorology, and we are looking at long-term timeseries and climatological statistics. For the region of Siberia, we consider only months from May to October for each year, given the lack of data coverage in winter months in this region due to snow-covered surfaces not being retrieved over.

### 3.2 Ground-based AOD and Angstrom Exponent (AE) from AERONET

We compare the Ref-D1 experiment to the column spectral AOD observations from AERONET. AERONET is a federated global network of ground-based sun photometers that measure total column aerosols properties (Holben et al. 1998). We consider the cloud-screened quality assured (Version 3 Level 2) monthly AOD and Armstrong Exponent (AE) (Giles et al. 2019). AE conveys information about the spectral variability in AOD, which itself is an indicator of particle size, where large values of AE (>1, indicating strong wavelength dependence) are generally associated with small particles (sub-micron) and small values (<1, indicating more spectrally flat AOD) are more indicative of coarse particles. For our climatological analysis, we consider all AERONET sites which have at least 10 months for each of the 12 climatological months over the period 2000-2018, similarly to what was done by (Gliß et al. 2021). For sites without direct measurements of AOD at 550 nm, AOD available at 500 nm or 532 nm and 667 nm or 675 nm are used to obtain AOD at 550 nm. For comparison with modelled AE at 470-870 nm, we use observations of AE at 440-870 nm. We obtain a total of 46 sites matching the selection criteria, and they are shown in Supplemental Figure 4 and listed in Supplemental Table 1. The model has been spatially sampled by co-locating the site coordinates with the nearest latitude-longitude model grid cell.





### 3.4 Surface PM₂.₅ from satellite-derived datasets

The modelled surface total PM$_{2.5}$ are compared with satellite-derived PM$_{2.5}$ model-data fusion datasets developed by the Atmospheric Composition Analysis Group at Washington University in St. Louis and available at
https://sites.wustl.edu/acag/datasets/surface-pm2-5/. In these datasets, monthly high-resolution surface PM$_{2.5}$ over land are estimated combining information from satellite derived AOD, a chemical transport model, and surface observations from ground-based monitoring networks. The AOD retrieved from multiple satellites is combined to produce a monthly best-estimate satellite-based AOD using a weighted average based on the level of agreement with AERONET. Simulations from the GEOS-Chem chemical transport model were used to establish the relationship between the total AOD and surface PM$_{2.5}$
and applied to the satellite AOD best estimate to produce a geophysical estimates of surface PM$_{2.5}$. Subsequent statistical fusion incorporated additional information and corrections from PM$_{2.5}$ ground-based measurements.

We use the dataset developed in van Donkelaar et al. 2021 at 0.1° × 0.1° resolution (dataset version V5.GL.03) to compare GEOSCCM simulated global and regional surface total PM$_{2.5}$. Monthly speciated PM$_{2.5}$ mass concentrations at 0.01° x 0.01° are provided for North America in separate dataset (van Donkelaar et al. 2019), which we use to compare the modeled total
PM$_{2.5}$ and components over the contiguous United States (CONUS). We use percentage PM$_{2.5}$ composition (dataset version V4.NA.02) applied to total PM$_{2.5}$ mass (dataset version V4.NA.03), to ensure mass closure as recommended by the data provider.

The main uncertainties related to these satellite-derived hybrid PM$_{2.5}$ estimates are linked to AOD retrieval limitations, AOD to PM$_{2.5}$ assumed relationship in the GEOS-Chem model, sparse ground monitoring in certain areas, and subgrid-scale
features. Monthly per-pixel uncertainties range from ~20% in populous Asian regions to 30-50% in North America but decrease significantly when aggregating data, making these PM$_{2.5}$ estimates reliable for regional-scale assessments despite pixel-level limitations.

To be compared with the model, the hybrid datasets have been spatially regridded to match the Ref-D1 spatial resolution. We also exclude areas where the hybrid dataset has no data in the monthly means at high latitudes (latitude > 68°).


### 3.5 Speciated PM₂.₅ and scattering coefficient from IMPROVE

We compare simulated speciated surface PM$_{2.5}$ concentrations and surface aerosol scattering coefficient with long-term observations over the US from the Interagency Monitoring of Protected Visual Environments Network mainly located in US national parks (Jenny L. Hand, et al. 2023). The IMPROVE network provides measurements of total PM$_{2.5}$ and of speciated
components (dust obtained from elemental species data, sea salt, OA, BC, nitrate, and sulfate). Filter samples are collected for 24h consecutive every 3 days. Collected PM mass is subsequently analyzed at controlled laboratory "dry" RH (30%-40%) conditions. For our climatological analysis of PM$_{2.5}$, we calculate monthly means from all IMPROVE sites that had at least 80% of measurements data with 'valid' status flag over the period 2000-2018 for total PM$_{2.5}$ and all its components (>1850



data points per station). We obtained a total of 101 sites matching the selection criteria, and they are shown in Supplemental
Figure 5 and listed in Supplemental Table 2. We sampled the model output by co-locating the site coordinates with the nearest
latitude-longitude model grid cell. From averaging modelled and observed PM$_{2.5}$ and individual components across all the
selected sites, we obtain average modelled and observed monthly speciated PM$_{2.5}$ across the US, which we use for our surface
PM$_{2.5}$ comparison. For the comparison of PM$_{2.5}$ speciated mass, we apply at a post-processing stage a growth factor correction
with relative humidity at RH=35% as defined in the GEOS documentation (Collow A. 2023) to match the IMPROVE PM
sampling conditions.

Additionally, we use long-term co-located measurements of speciated PM$_{2.5}$ and scattering coefficient to investigate the
relationship between surface PM$_{2.5}$ and surface aerosol optical properties. Scattering coefficient at 550 nm together with
ambient RH is measured at a subset of IMPROVE sites through open-air integrating nephelometers. Measurements are reported
hourly. We filter hourly nephelometer measurements with RH <=95%, with scattering coefficient < 5000 m$^{-1}$ and rate of
change less than 50 Mm$^{-1}$h$^{-1}$ to exclude the interference of meteorology (e.g. fog episodes), following the procedure used in
(Latimer and Martin 2019). Subsequently, we consider only days with at least 12 valid hourly measurements to obtain average
daily values. We then consider stations with co-located scattering and PM$_{2.5}$ measurements and subset only days where both
valid PM$_{2.5}$ and scattering are measured. We select months with at least 6 valid daily measurements out of a max of 11 (PM
are measurements are done every 3 calendar days). This selection procedure results in 6 sites with long-term measurements of
both speciated PM$_{2.5}$ and scattering coefficient and RH, presented in Supplemental Figure 6 and listed in Supplemental Table
3.

**3.6 Reconstruction of surface scattering from model diagnostics and IMPROVE observations**

We use the long-term co-located measurements of speciated PM$_{2.5}$ and scattering coefficient to investigate the relationship
between surface PM$_{2.5}$ concentrations and aerosol optical properties. Total surface scattering coefficient of an aerosol
population $\sigma_{sca}$ [m$^{-1}$] is a function of mass concentration, size distribution, optical properties, and local relative humidity for
each aerosol component in the population. In GOCART, the total scattering coefficient $\sigma_{sca}$ of an aerosol population of $k$
species is calculated assuming externally mixed aerosol as:

$$sca = \sum_{n=1}^{k} \left( b_{sca,n}(LUT_n, RH) * m_n \right) \qquad\qquad 1$$


where $b_{sca,n}$ is the mass scattering efficiency for each aerosol species $n$, which depends on the model LUTs that contain the
assumed information on aerosol optical properties and size distribution and their dependence on RH. $m_n$ is the dry mass
concentration for each aerosol species $n$. The $k$ species considered in GEOSCCM Ref-D1 in the calculation are organic carbon,
black carbon, nitrate, sulfate, dust and sea salt.





Following from the approach developed in (Latimer and Martin 2019), we compare the observed $\sigma_{sca}$ from IMPROVE with the reconstructed scattering $\sigma_{sca,RE}$. We reconstruct the scattering coefficient in Equation 1 using GEOSCCM LUTs for 3 different combinations of modeled and observed RH and $PM_{2.5}$ composition to isolate the importance of modeled LUT, RH and $PM_{2.5}$ composition respectively in simulating $\sigma_{sca}$ (Section 4.4). As indicated in Section 3.5, the IMPROVE optical measurements are at ambient RH, while $PM_{2.5}$ and $PM_{10}$ measurements are at a controlled, low RH comparable to dry

conditions.

    One limitation of our approach is that we use fine mode speciated mass to reconstruct total (fine+coarse) observed scattering coefficient, because of the lack of IMPROVE observations on coarse mode. This could lead to underestimating the reconstructed scattering. On the other hand, nephelometers are affected by a truncation error, which is a systematic error increasing as the particle size increases, thus scattering from the coarse mode can be underestimated (Molenar J. V. 1997). To

reduce the uncertainty in determining the scattering coefficient for both the observed and reconstructed scattering, we restrict our analysis to conditions dominated by fine-mode aerosols i.e. when IMPROVE observed $PM_{2.5}/PM_{10} > 0.5$.

**Table 1: summary table of aerosols observations used for comparison with the GEOSCCM Ref-D1 simulation.**

| observation | platform | time coverage | spatial coverage | source | reference |
|---|---|---|---|---|---|
| Aerosol Optical Depth | satellite | 2003-2018 | global | MODIS Aqua (NNR retrieval) | (Randles et al. 2017) |
| Aerosol Optical Depth | ground-based | 2000-2018 | global | AERONET | (Holben et al. 1998) |
| Angstrom Exponent | ground-based | 2000-2018 | global | AERONET | (Holben et al. 1998) |
| Surface total $PM_{2.5}$ mass | data-model fusion | 1998-2018 | global | Washington U. S. Louis | (van Donkelaar et al. 2021) |
| Speciated surface $PM_{2.5}$ mass | data-model fusion | 2000-2016 | US | Washington U. S. Louis | (van Donkelaar et al. 2019) |
| Speciated surface $PM_{2.5}$ mass | ground-based | 2001-2018 | US | IMPROVE network | (Jenny L. Hand, et al. 2023) |
| Surface aerosol scattering coefficient | ground-based | 2001-2018 | US | IMPROVE network | (Jenny L. Hand, et al. 2023) |

## 4 Results

**4.1 Global column AOD**

    Figure 4 shows the comparison of regional mean monthly timeseries and mean annual cycle of AOD between Ref-D1 and MODIS NNR retrievals. The global climatological monthly spatial distribution is reported in Supplemental Figure 7. GEOSCCM reproduces month-to-month variability of AOD over land for all regions ($0.6 < r < 0.9$) except in Europe and

Siberia ($r \simeq 0.3$). However, there are strong regionally and seasonally varying biases evident. GEOS CCM simulated AOD is biased low by 38%-58% in biomass burning regions of Southern Africa, North America, South America, Siberia during peak fire months (Figure 4b). For South America and Southern Africa, simulated AOD is underestimated throughout the entire year.



Over Northern Africa, the model underestimates AOD during the spring season and overestimates the seasonal summertime and early autumn peak in magnitude (within ~30%) and shows a delay of 1-2 months in the peak of AOD compared to

observations, with bias mainly concentrated over the South Sahara steppe and woodlands (Supplemental Figure 7). In Southern Asia the model shows a good agreement with observations in all seasons except winter, when AOD is overestimated (+26%). A persistent positive bias is found over Eastern China almost throughout the year (Supplemental Figure 7). Over Europe the model has a high bias in AOD in all seasons (between +36% and + 64%) except summer, when the AOD is underestimated (-13%). Finally, over Australia, the model tends to overestimate AOD in summer (+35%) and slightly underestimate AOD over

winter (-13%), but the overall seasonal variability is well reproduced.

We find consistent results with the comparison with satellite retrieved AOD when comparing the modeled total column AOD with observations at selected AERONET sites (Supplemental Figure 8). GEOSCCM AOD is on average underestimated in biomass burning influenced regions and overestimated over Europe, and GEOSCCM overestimates AOD at sites over North America during all seasons except summer (i.e., peak biomass burning season). However, at sites over Asia in India and China

the model underestimates AOD compared to AERONET, which is somewhat at odds with the results obtained with the comparison with satellited retrieved AOD, for which modeled AOD was overestimated. Total column AE tends to be globally underestimated in the model compared to AERONET observations (Supplemental Figure 9), suggesting that the model simulates either too much coarse aerosol or not enough smaller aerosols, as also found on average in the AeroCom multi-model evaluation (Gliß et al. 2021). An exception are Asian sites and central Brazil, where AE is overestimated. Additional

climatological comparison and statistics of AOD and AE for each selected AERONET site are presented in Supplemental Figure 10, 11 and Supplemental Table 4.





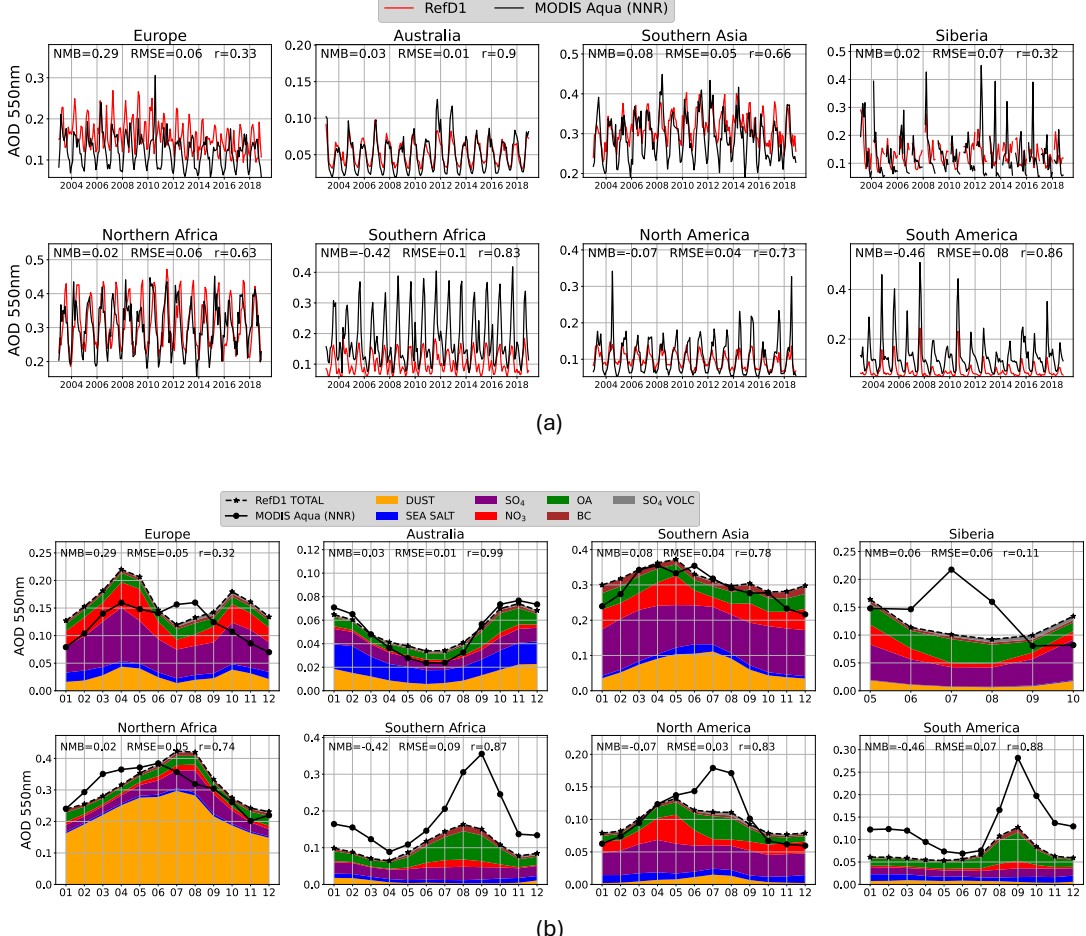

**Figure 4: Comparison of GEOSCCM Ref-D1 and MODIS Aqua (a) regional annual timeseries and (b) regional seasonal cycle of AOD for 2003-2018. For GEOSCCM, the AOD seasonal cycle is reported as speciated. Siberia seasonal cycle is from May to October only.**






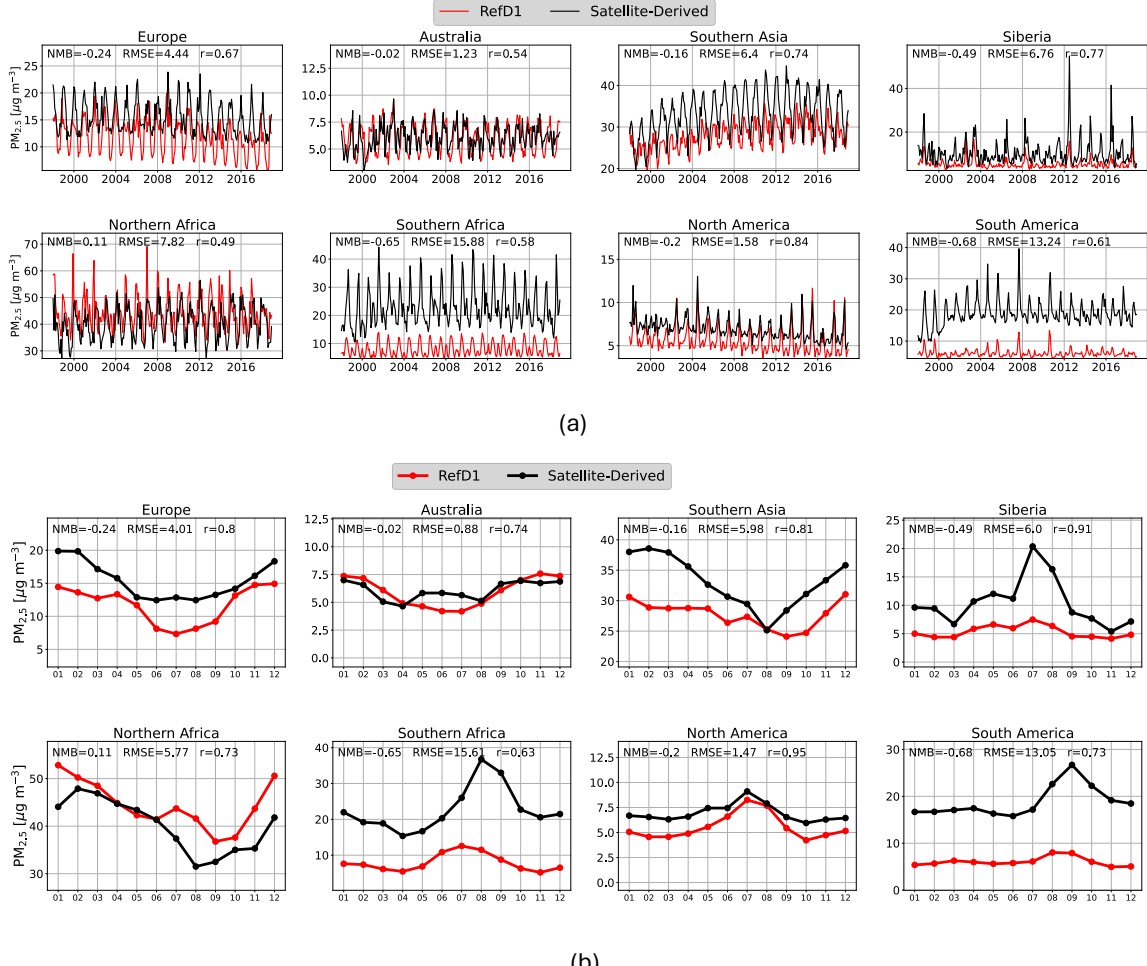

**Figure 5: Comparison of GEOSCCM Ref-D1 (red) and satellite derived (black) PM2.5 for the period 1998 – 2018. Shown are (a) regional annual timeseries and (b) regional annual cycle. Note the y-axes do not necessarily start at zero.**


## 4.2 Global surface total PM₂.₅ concentrations

Comparisons of the GEOSCCM Ref-D1 simulated PM₂.₅ concentrations with satellite-derived PM₂.₅ show similarities with the comparison with total column AOD. Figure 5 shows the regional mean monthly timeseries and mean annual cycle comparison for surface PM₂.₅ concentrations. The global climatological monthly spatial distribution is reported in

Supplemental Figure 12. The model captures the month-to-month variability in regional PM₂.₅ compared to the satellite-





derived PM$_{2.5}$ in all regions (r > 0.5), especially in Southern Asia, North America, and Siberia (0.73 < r < 0.84). However, the modeled PM$_{2.5}$ concentrations are biased low in all regions throughout the seasonal cycle, except for Northern Africa, which has a small positive bias, and Australia, which has a mixed bias.

Similar to AOD, the model underestimates PM$_{2.5}$ in biomass burning regions, but does overall reproduce the seasonal cycle, except for Southern Africa, where the model predicts the peak season to occur one month earlier than compared to observations. During peak fire season, highest biases are found in South America and Southern Africa (~70%) and Siberia (~60%), while the model performs better in North America (~ -10%). Over Northern Africa, the model overestimates PM$_{2.5}$ by around 20% in all seasons except spring. For Southern Asia, the model underestimates PM$_{2.5}$, especially during wintertime (-20%), while in Europe the bias is largest in the summertime (-37%). Model bias is largest in Australia during the southern

hemisphere winter (20%). Notably, over Europe there is a much better correlation of the climatological modeled PM$_{2.5}$ with observations (r = 0.8) than for AOD (r = 0.33), suggesting some disconnect between the modeled mass at the surface and the aerosols optical properties at the column level and or some uncertainties related to the transport in the model.

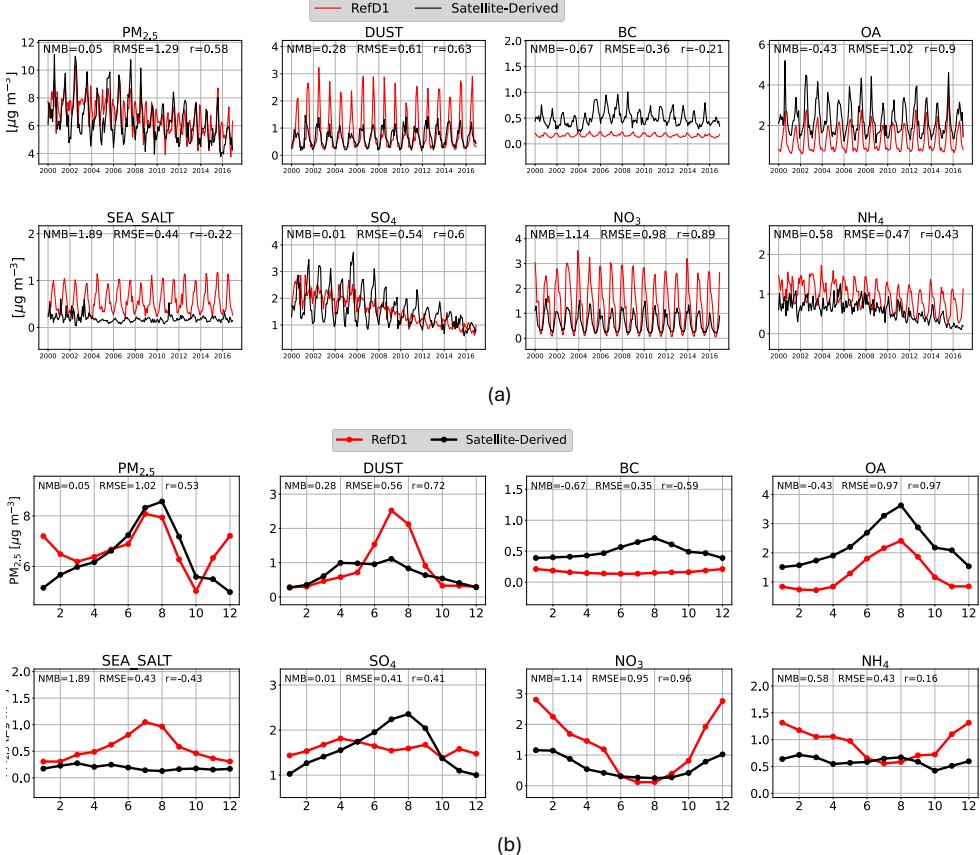

**Figure 6. Comparison of speciated PM$_{2.5}$ over CONUS between GEOSCCM (REFD1, red) and satellite-derived PM$_{2.5}$ (black) at RH=35%. (a) Annual timeseries and (b) mean seasonal cycle for the period 2000-2003.**



### 4.3 PM$_{2.5}$ composition over CONUS

We use both satellite-derived and ground-based observations of speciated PM$_{2.5}$ over CONUS to investigate in more detail the GEOSCCM ability to represent aerosol composition at the surface. Figure 6 shows the comparison of speciated surface PM$_{2.5}$ concentration timeseries and seasonal cycle between GEOSCCM and satellite-derived data. The spatial distribution comparison is reported in Supplemental Figure 13. Figure 7 shows the comparison of speciated surface PM$_{2.5}$ concentrations timeseries and spatial distribution between GEOSCCM and ground-based observations from the IMPROVE network.

GEOSCCM has a similar decreasing trend in total PM$_{2.5}$ as in the satellite-derived observations (Figure 6), although it has
a positive bias during wintertime. The performance for individual species varies. GEOSCCM Ref-D1 reproduces the decreasing trend in SO$_4$ both for ground-based observations and satellite-derived SO$_4$, and to a lesser extent the decreasing trend in NH$_4$ (available for satellite-derived data only). However, the model is not able to fully capture the SO$_4$ and NH$_4$ seasonal cycle and is positively biased especially in winter months. The comparison with both satellite-derived data and ground-based observations shows an average monthly positive bias in natural aerosols of sea salt (Normalized Mean Bias
NMB $\simeq$ 2 and NMB $\simeq$ 12 respectively) and dust (NMB $\simeq$ 0.3 and NMB $\simeq$ 0.1 respectively). This is most pronounced in the summer, and may be linked to excessive inland intrusion of dust and sea salt mainly from the Atlantic Ocean (Figure 7b and Supplemental Figure 13). The model shows a high positive bias in nitrate during the winter when compared both to ground-based observations (NMB = 3.61 and satellite-derived data NMB = 1.14), driving an overall positive bias in total PM$_{2.5}$ in the winter, which is also affected by model-overestimated SO$_4$ and NH$_4$. On the other hand, GEOSCCM is skillful in reproducing
OA interannual variability (r = 0.9) and capture better magnitude compared to the ground-based observations (NMB $\simeq$ -0.12) than to the satellite derived estimates (NMB$\simeq$ -0.43). For BC, the model has also a smaller magnitude bias against the ground-based observations than for satellite-derived ones (NMB = 0.03 vs NMB = -0.67 respectively) but is not able to capture the month-to-month variability in either (r < ~0.2). Despite the described biases, the model captures the spatial distribution of observations for most PM$_{2.5}$ components, except for sea-salt and dust as already noted and for nitrate (Figure 7b).

The differences between the comparison of GEOSCCM with the satellite derived PM$_{2.5}$ and the ground-based monitors may arise from the fact that IMPROVE ground-based sites are representative of background concentrations, being sites located in national parks across the US, while the satellite-derived PM$_{2.5}$ are representative of different chemical environments at higher resolution (background, rural, urban), which the resolution of current GEOSCCM is not expected to capture. This is evident for example in the performance of BC, which has a smaller bias in the comparison with observations from the
IMPROVE sites than the satellite-derived ones. In addition, this comparison could be sensitive to different assumptions in the AOD and PM$_{2.5}$ relationship in GEOSCCM versus the one assumed in the GEOS-Chem model which is used to obtain the satellite-derived PM$_{2.5,}$ especially in areas where the incorporated additional information and corrections from PM$_{2.5}$ ground-based measurements is sparse.








(a)                                                    (b)

**Figure 7. (a): Ref-D1 modelled at RH=35% (red, Ref-D1) vs observed (black, IMPROVE) monthly mean timeseries of**
**speciated PM$_{2.5}$, averaged over the observation sites throughout CONUS. Lines represent the median, while shading**
**encompasses the 25$^{th}$ -75$^{th}$ percentiles. (b) Modelled vs observed (circles, IMPROVE) yearly average of surface PM$_{2.5}$**
**and composition over CONUS (2001-2018).**




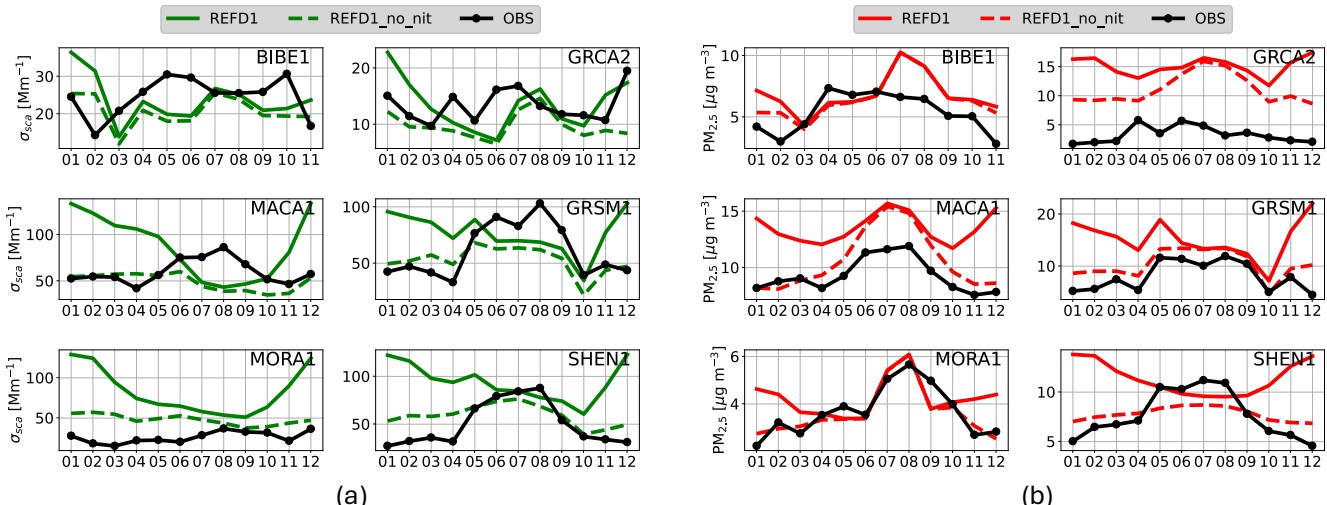

**Figure 8. Seasonal cycle of (a) surface scattering coefficient and (b) total PM2.5 at selected IMPROVE sites. Black lines represent IMPROVE observations, solid lines are Ref-D1 and dashed lines are Ref-D1 with the nitrate component removed.**


**Table 2: statistics referring to Figure 8.**

| station | run | σsca | | | PM2.5 | | |
|---|---|---|---|---|---|---|---|
| | | NMB | RMSE [Mm-1] | r | NMB | RMSE [ug m-3] | r |
| BIBE1 | REFD1 | -0.09 | 10.27 | -0.19 | 0.29 | 2.67 | 0.31 |
| BIBE1 | REFD1_no_nitrate | -0.18 | 10.20 | -0.14 | 0.26 | 2.54 | 0.38 |
| GRCA2 | REFD1 | -0.03 | 6.14 | 0.22 | 3.23 | 11.85 | 0.07 |
| GRCA2 | REFD1_no_nitrate | -0.23 | 6.41 | 0.19 | 2.38 | 9.16 | 0.24 |
| GRSM1 | REFD1 | 0.08 | 37.25 | 0.22 | 0.60 | 7.12 | 0.38 |
| GRSM1 | REFD1_no_nitrate | -0.18 | 30.66 | 0.60 | 0.23 | 3.14 | 0.84 |
| MACA1 | REFD1 | 0.50 | 57.50 | -0.08 | 0.47 | 5.04 | 0.59 |
| MACA1 | REFD1_no_nitrate | -0.17 | 28.77 | 0.26 | 0.14 | 2.79 | 0.71 |
| MORA1 | REFD1 | 1.73 | 54.27 | -0.17 | 0.04 | 1.90 | 0.28 |
| MORA1 | REFD1_no_nitrate | 0.80 | 27.64 | -0.18 | -0.03 | 1.79 | 0.40 |
| SHEN1 | REFD1 | 0.91 | 60.89 | 0.00 | 0.51 | 5.99 | -0.25 |
| SHEN1 | REFD1_no_nitrate | 0.19 | 26.38 | 0.59 | 0.02 | 2.96 | 0.70 |




**4.4 Connecting surface PM$_{2.5}$ mass and scattering coefficient**

Differences in the comparisons of AOD and PM$_{2.5}$ to observations prompt us to investigate the relationship between optical properties and aerosol mass in the model. For this, we use co-located IMPROVE measurements of speciated PM$_{2.5}$ and the scattering coefficient at selected sites, as described in Section 3.5. We find that the model has a better performance in
reproducing surface total PM$_{2.5}$ than scattering coefficient for the selected sites (Figure 8, Table 2). Although the bias in the scattering coefficient and PM$_{2.5}$ varies across sites, the model is not able to capture the seasonal variability of the scattering coefficient at all sites (-0.2 < r < 0.2), while it does better for the seasonality of PM$_{2.5}$ (-0.25 < r < 0.59). The performance of the model improves significantly both for the scattering coefficient and the total PM$_{2.5}$ when removing the nitrate component, which we found is largely overestimated in the model in winter over CONUS (Section 4.3), although it is still better for PM$_{2.5}$
(0.24 < r < 0.84) than for the scattering coefficient (-0.18 < r < 0.6).

We reconstruct the scattering coefficient ($\sigma_{sca,RE}$) for different combinations of modeled and observed RH, PM$_{2.5}$ and LUT (Section 3.6) and compare it with the observed scattering coefficient ($\sigma_{sca}$) to help understand the role of each of these factors in the disconnect between the model performance for the scattering coefficient and PM$_{2.5}$. For each species *n* these combinations are: 1) $[LUT_{mod}, RH_{obs}, m_{obs}]$ to isolate the impact of GEOSCCM assumed optical tables; 2) $[LUT_{mod}, RH_{mod},$
$m_{obs}]$ to isolate the impact of simulated relative humidity; and 3) $[LUT_{mod}, RH_{obs}, m_{mod}]$ to isolate the impact of simulated PM$_{2.5}$ composition.

Figure 9 shows the comparison between the reconstructed and observed scattering coefficient. We find that the $\sigma_{sca,RE}$ using the model LUT with the observed RH and PM$_{2.5}$ (Figure 9, blue line) is in good agreement with the observed $\sigma_{sca}$ with generally the lowest NMB and RMSE and highest correlation for this case, indicating that the combination of optical properties
and size distribution assumptions in the LUT are generally compatible with observations. By contrast, the reconstructed $\sigma_{sca,RE}$ that uses the model LUT, observed PM$_{2.5}$ and modeled RH (Figure 9, green line) exhibits a consistent negative bias compared to observations for all sites. This is expected since the model-simulated RH is generally lower than observed at all sites (Figure 10b) and the scattering coefficient increases as RH increases. The results are more ambiguous for $\sigma_{sca,RE}$ using the model LUT, observed RH, and modeled PM$_{2.5}$ (Figure 9, orange line) suggesting that the modeled composition plays a more complicated
role in affecting the simulated scattering coefficient. At the selected sites, beside the high bias in nitrate, GEOSCCM Ref-D1 is not able to fully capture the observed PM$_{2.5}$ seasonal composition (Figure 10a), with biases for each component in line with what is found for the comparison of speciated PM$_{2.5}$ averaged over CONUS in Section 4.3. However, when removing the nitrate component and reconstructing the scattering, the performance of the model improves (Supplemental Figure 14, Supplemental Table 5).






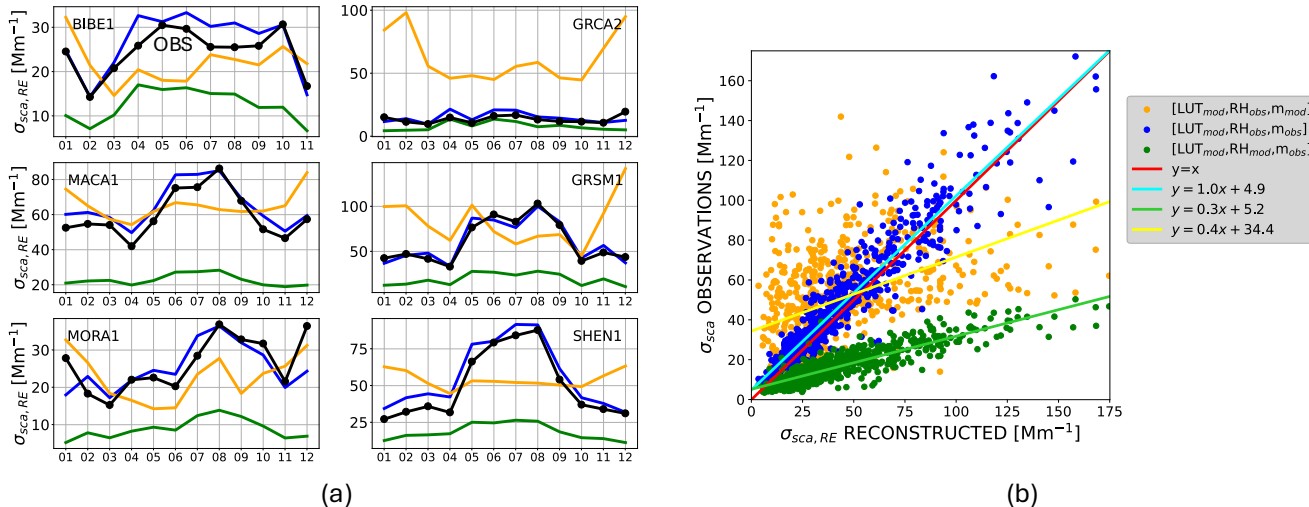

465                                                                              (a)                                                                                           (b)

**Figure 9. Reconstructed scattering for 3 different combinations of modelled and observed RH, PM$_{2.5}$ and LUT as described in Section 3.6. (a) Seasonal cycle of reconstructed scattering at the selected sites. Observations are reported with the black line. (b) Scatter plot of the reconstructed scattering with all monthly data.**


**Table 3: metrics for the reconstructed scattering $s_{sca,RE}$ compared to IMPROVE scattering observations for the three combination of model and observations considered: 1) $[LUT_{mod}, RH_{obs}, m_{obs}]$, 2) $[LUT_{mod}, RH_{mod}, m_{obs}]$ , 3) $[LUT_{mod}, RH_{obs}, m_{mod}]$.**

| station | NMB 1 | NMB 2 | NMB 3 | RMSE 1 [Mm-1] | RMSE 2 [Mm-1] | RMSE 3 [Mm-1] | r 1 | r 2 | r 3 |
|---------|-------|-------|-------|---------------|---------------|---------------|-----|-----|-----|
| BIBE1 | 0.12 | -0.48 | -0.16 | 5.18 | 13.52 | 9.21 | 0.84 | 0.64 | 0.06 |
| GRCA2 | 0.14 | -0.37 | 3.25 | 5.73 | 6.73 | 46.71 | 0.67 | 0.58 | 0.31 |
| GRSM1 | 0.00 | -0.68 | 0.1 | 13.97 | 54.85 | 40.28 | 0.92 | 0.86 | 0.16 |
| MACA1 | 0.08 | -0.62 | 0.1 | 12.09 | 41.45 | 23.08 | 0.9 | 0.86 | 0.48 |
| MORA1 | 0.02 | -0.62 | -0.22 | 9.85 | 20.02 | 15.13 | 0.71 | 0.62 | 0.24 |
| SHEN1 | 0.13 | -0.63 | 0.1 | 11.55 | 38.51 | 31.6 | 0.95 | 0.9 | 0.07 |




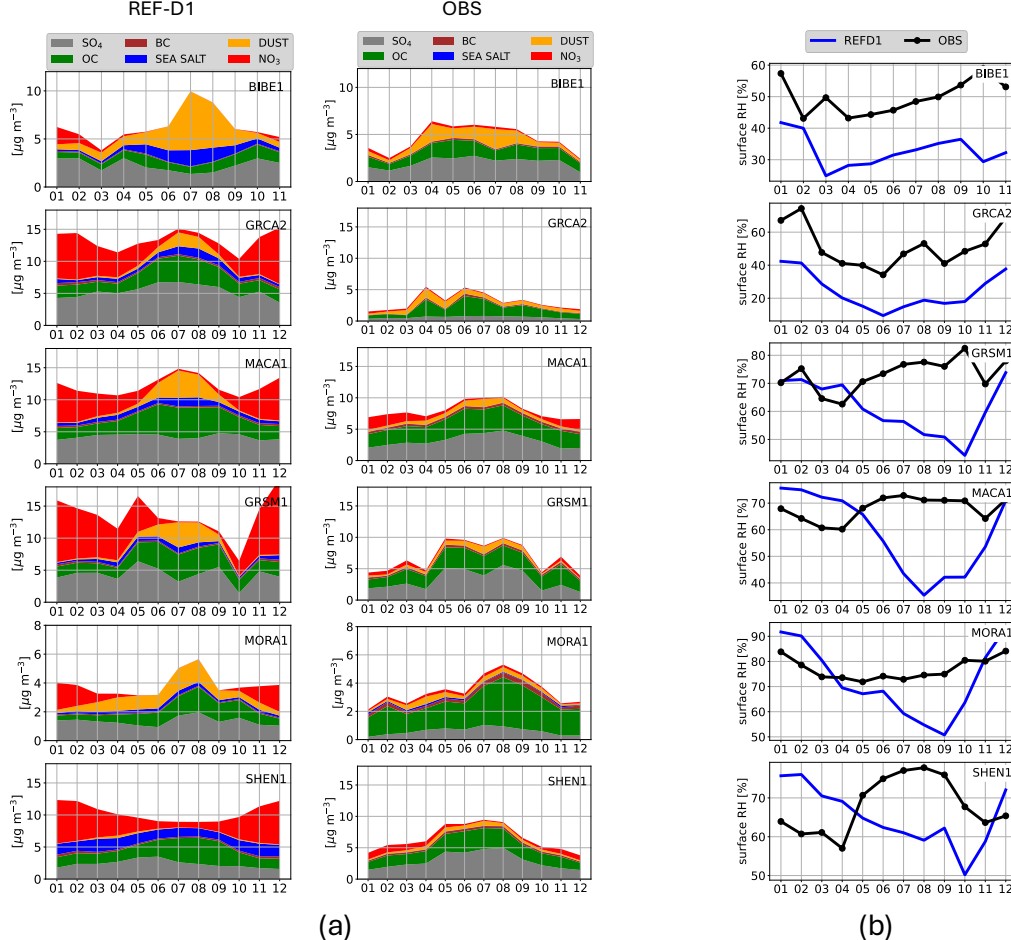

(a)                                                                    (b)


**Figure 10. Comparison between Ref-D1 and observations for the seasonal cycle of (a) surface PM$_{2.5}$ components (a) and (b) surface RH at selected IMPROVE sites.**



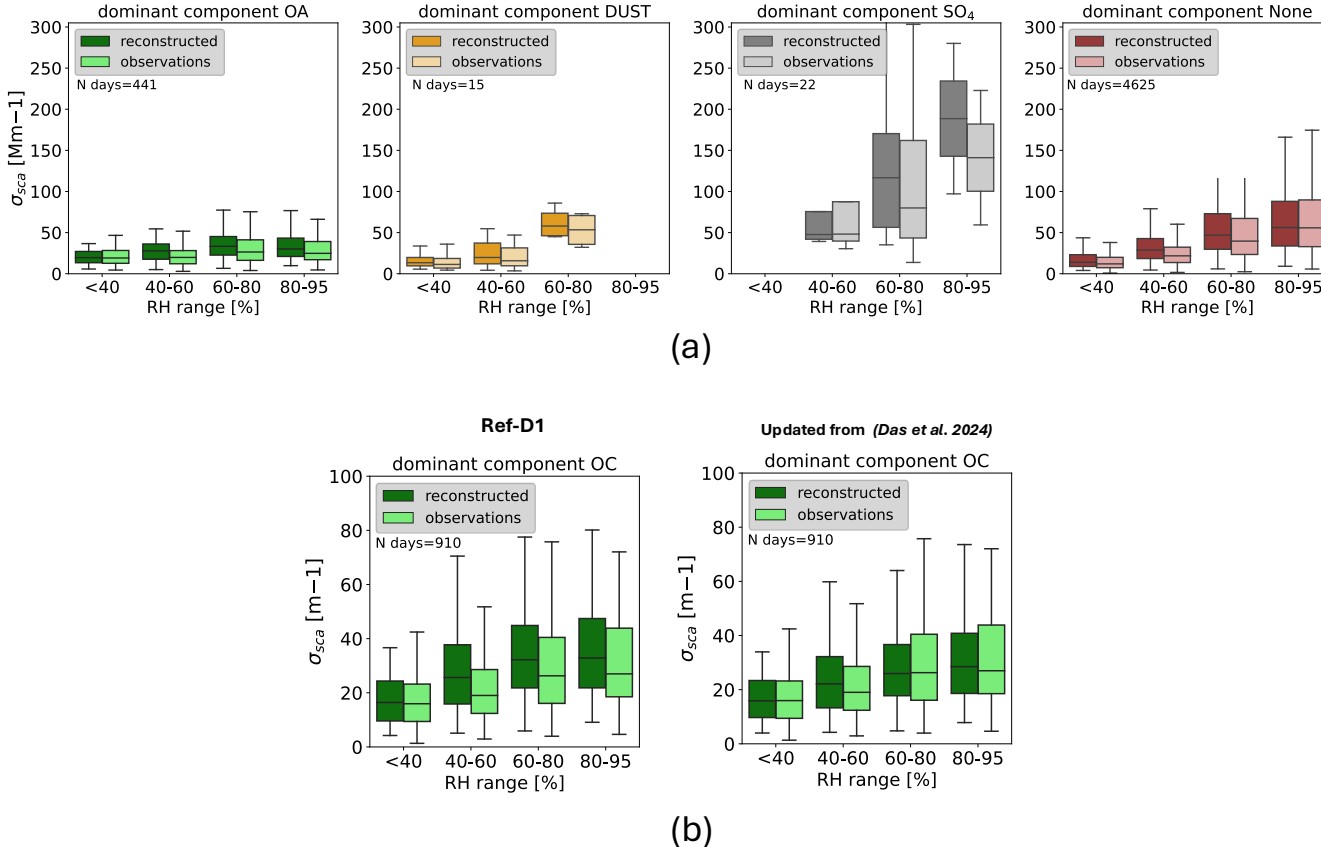

**Figure 11. (a) Comparison of the observed and reconstructed scattering coefficient for different RH ranges and different dominant component in the PM$_{2.5}$ (fraction > 0.5) using daily IMPROVE observations of speciated PM$_{2.5}$, RH and scattering coefficient. (b) Dame as (a) but with reconstructed scattering with updated LUT for OC.**

We further analyze the role of model assumptions of aerosol optical properties by sorting the reconstructed scattering coefficient for different RH range and dominant PM$_{2.5}$ component, as shown in Figure 11. Overall, where the dominant component is OA, dust, or SO$_4$, the distribution of the reconstructed scattering coefficient agrees well with observations for all different RH ranges, suggesting that the model assumptions of aerosol optics for each of these species is adequate. However, it is possible to further improve the model representation of aerosol properties. For example, the reconstructed scattering coefficient derived using on an updated set of optical properties developed by Das et al. (2024) for biomass burning organic aerosol, is in better agreement with observed scattering coefficients compared to using default optics, especially in regions where OA is the dominant component (Figure 11b).



## 5 Discussion and Conclusions

In this study we assessed GEOSCCM capabilities and limitations in simulating aerosols by focusing on the link between
aerosol optical properties and mass, examining key factors that contribute to uncertainties in simulated long-term variability, regional and seasonal variations of AOD and PM$_{2.5}$, and the consistency of GEOSCCM PM$_{2.5}$ simulation with optical properties, including the role of simulated RH on hygroscopic scattering enhancement. This work also presents the first extensive comparison between the GEOSCCM aerosol component and observational data.

Our results show that the GEOCCM reproduces month-to-month variability and long-term trends of total column AOD
and surface PM$_{2.5}$ concentrations over most land regions, except Europe. However, GEOSCCM shows regional biases in AOD and PM$_{2.5}$ that show similar global patterns. In particular, GEOSCCM underestimates both AOD and PM$_{2.5}$ in biomass burning regions and overestimates dust over the Sahara Desert. Performance for speciated PM$_{2.5}$ varies. Specifically, the GEOSCCM model overestimates secondary inorganic PM$_{2.5}$ concentrations across the continental United States, with nitrate showing the most significant overestimation. Additionally, the model produces excessive values for sea-salt in coastal regions and dust
during summer months. The model performs better for carbonaceous species (OA and BC). Biases in the magnitude of individual PM$_{2.5}$ components also have a significant impact in the simulated aerosol scattering, as does the model-simulated relative humidity, which is generally lower than observed, while we find that the scattering efficiency assumptions in GEOSCCM are consistent with observations.

One cause of aerosol loading biases in biomass burning regions can be related to biases in the emission inventory used in
the GEOSCCM Ref-D1 simulation, which is based on fire emissions estimations from burned-area estimates (Section 2.2). Burned-area driven fire inventories have been shown to give lower overall biomass burning aerosol emissions and result in low AOD over biomass burning regions in GEOS compared to emissions derived from fire radiative power, which are by design constrained by observed satellite AOD, as well as use different set of emissions factors (Pan et al. 2020). This is further confirmed by comparing the simulated AOD by GEOSCCM with the one simulated with a recent benchmark run of GEOS,
GOCART-2G run, (Collow et al. 2024) but using a fire radiative power-based inventory (QFEDv2.5 (Darmenov, A. and da Silva, A.: 2015)). The GOCART-2G run estimates higher AOD over biomass burning region than GEOSCCM and is closer to observations especially during peak fire season, since by design it is specifically tuned to produce a reasonable observed AOD. While increasing the emissions of biomass burning would improve GEOSCCM performance in simulating AOD and PM$_{2.5}$ mass over relevant regions, there might be additional reasons that contribute to the low bias, such as not utilizing a plume-rise
parametrization (Ke et al. 2021), incorrect lifetime from incorrect precipitation simulation, and particle size-distribution assumptions that affect the assumed mass scattering efficiencies (Zhong et al. 2022).

For the overestimation in GEOSCCM of natural aerosols of dust and sea salt there might also be multiple factors at play. GEOSCCM dust summer peaks might be higher than observations because the current model does not include seasonal vegetation cover changes that likely suppress dust emissions, especially over Southern Saharan steppe and woodlands in
summer. While both for sea salt and dust, the model coarse resolution might lead to more inland intrusion of these species due



to higher numerical diffusion over coastal regions as found for the regional PM$_{2.5}$ composition analysis over CONUS. Additionally, the internal resolution-dependent dust emission scaling factor used may have been inappropriate. Finally, the free running meteorology in the GEOSCCM configuration can introduce differences in the long-range transport dynamics and removal processes compared to a run using replayed or assimilated meteorology. As shown in Supplemental Figure 15 GEOSCCM is relatively more efficient at transporting dust from the Sahara across the North Atlantic than the GEOS GOCART2G run performed with the MERRA-2 assimilated meteorology (Collow et al. 2024).

Analysis of the speciated PM$_{2.5}$ over CONUS highlights that GEOSCCM overall can reproduce long-term trends of total and speciated PM$_{2.5}$ and their spatial distribution features, but there are biases in individual components. Besides the biases in biomass burning and natural aerosols already discussed, GEOSCCM shows a positive bias in secondary inorganic components (NO$_3$, SO$_4$ NH$_4$), especially during wintertime. The nitrate aerosol component shows the greatest positive bias, which also drives a high positive bias in total PM$_{2.5}$ during winter. One main reason for the bias in nitrate can be related to potential problems in nitric acid (HNO$_3$) mass closure in the coupled GMI-GOCART scheme in GEOSCCM. As described in Section 2.2, nitric acid chemistry is computed in GMI and then passed as an input to GOCART for nitrate chemistry computation. However, nitric acid loss due to nitrate formation is not fed back to GMI, and likely results in an overestimation of total HNO$_3$ available. Indeed, when using a configuration with decoupled chemistry-aerosols using scaled HNO$_3$ fields as input to GOCART, the GEOS model has a better performance for nitrate (Collow et al. 2024). In addition, the positive bias in natural aerosols provides more surface for heterogeneous production of nitrate onto sea salt and dust particles, which is the main pathway for nitrate aerosol formation (Huisheng Bian et al. 2017), leading to potential overestimation of nitrate formed through heterogenous chemistry.

The analysis of co-located scattering coefficient, speciated PM$_{2.5}$ and reconstructed scattering coefficient shed lights over the uncertainties driving the link between aerosol mass and aerosol optical properties in current GEOSCCM. We found that current optics LUT assumptions used in the model are overall compatible with observations. However, aerosol optical properties can be further improved, as shown for the biomass burning emitted OA example. Extending the assessment of GEOSCCM assumptions of aerosol optical properties beyond the CONUS will be also desirable, in order to increase the confidence in optics LUT on a global scale spanning different physio-chemical environments. The main uncertainties in the aerosol mass-optical properties relationship are driven by model-simulated relative humidity and model simulated PM$_{2.5}$ composition, which play an important role in determining biases in simulated surface scattering coefficient. In particular, model-simulated RH is generally lower than observed over CONUS, driving a constant negative bias in simulated scattering coefficient, that could be linked to deficiencies in our methodology in deriving surface RH accurately enough with free-running meteorology. It has also been shown that model calculated AOD is sensitive to the spatial and temporal resolution of atmospheric relative humidity, due to the highly non-linear relationship between RH and the aerosol mass extinction efficiency (H. Bian et al. 2009). The bias introduced by the modeled aerosol composition has a more mixed effect in affecting the simulated scattering coefficient (both positive and negative biases), which can be more complex to disentangle. Nevertheless,





improving the model's ability to correctly simulate aerosol composition would improve the simulated scattering coefficient,
as shown in the experiment with nitrate removed.

These findings serve as a benchmark for GEOSCCM aerosol improvements, with the following priorities to reduce uncertainties in the aerosols mass, optical properties and their relationship: 1) aerosol mass loading and optical properties can both be improved in GEOSCCM, possibly by using a fire radiative power based inventory fire emissions which will reduced seasonal biases in biomass burning regions; 2) biases in $PM_{2.5}$ and surface scattering coefficient can be reduced by reducing
biases in nitrate mass loading component, which require further investigation in the coupled GMI-GOCART module and in the nitrate heterogeneous production onto natural aerosols surface; 3) and accurate model representation of RH could substantially improve the representation of the aerosol scattering coefficient.

The challenges identified in GEOSCCM's aerosol representation reflect issues that are present more broadly in the context of aerosol modeling. Our identified RH biases reflect a persistent challenge across Earth system models (Dunn et al. 2017). In
particular, our work highlights the implication of biases in the simulated relative humidity on the aerosol scattering coefficient, which is of importance for climate models where the meteorology is free running and not prescribed based on reanalysis as in chemistry transport models. This is in addition to the diversity in how models represent aerosol light scattering enhancement with humidity, emphasizing that hygroscopicity plays an important role in simulated aerosol extinction (Latimer and Martin 2019; Burgos et al. 2020). Assumptions about aerosol size distributions and incorrect mass extinction coefficients can also
significantly impact scattering calculations (Latimer and Martin 2019; Zhong et al. 2022).

Our work reinforces the importance of synergistic use of multiple co-located and coincident observations to effectively investigate and constrain modeled aerosols properties in climate models. By simultaneously assessing aerosol AOD, surface $PM_{2.5}$ mass concentration and composition, and surface scattering coefficient, it was possible to identify key sources of uncertainties in GEOSCCM's current aerosol representation and in the link between aerosol optical properties and mass. This
co-incident and co-located observational constraint approach provides a framework for model improvement that extends beyond traditional independent parameters evaluation, helping to clarify the interconnected properties and processes affecting aerosol representation in atmospheric models, and improve their subsequent applications.

**Code Availability**

GEOS, including the GEOSCCM configuration, is a publicly available Earth System model with source code
at https://github.com/GEOS-ESM.

**Data Availability**

All observational data used are from publicly available datasets.



MODIS Level 2 reflectances are available from https://doi.org/10.5067/MODIS/MOD04_L2.006 (Levy, R., Hsu, C., and et al. 2015a) for Terra and https://doi.org/10.5067/MODIS/MYD04_L2.006 (Levy, R., Hsu, C., and et al. 2015b) for Aqua.

AERONET observations can be downloaded at https://aeronet.gsfc.nasa.gov/cgi-bin/webtool_aod_v3.

IMPROVE network data can be downloaded from the Federal Land Manager Environmental Database at

http://views.cira.colostate.edu/fed/DataWizard/Default.aspx.

Satellite-derived $PM_{2.5}$ datasets are available at https://sites.wustl.edu/acag/datasets/surface-pm2-5/.

GOCART optical tables are available at https://portal.nccs.nasa.gov/datashare/iesa/aerosol/AerosolOptics/.

(Last access for all the data May 7, 2025).

**Author Contribution**

CM, PRC, conceived the study. PRC, QL, LH, KEN, VV, SAS, SS, MM participated in the set-up and run of the RefD1 experiment. CM led the software, data curation and analysis with contribution from PRC and ABC. CM was responsible for original draft preparation, and PRC, ABC, SAS, SD and QL contributed to review and editing.

**Competing Interests**

The authors declare no competing interests.

**Acknowledgements**

Simulations were performed at the NASA Center for Climate Simulation (NCCS).

IMPROVE is a collaborative association of state, tribal, and federal agencies, and international partners. US Environmental Protection Agency is the primary funding source, with contracting and research support from the National Park Service. The

Air Quality Group at the University of California, Davis is the central analytical laboratory, with ion analysis provided by Research Triangle Institute, and carbon analysis provided by Desert Research Institute.

We thank the AERONET (PI(s) and Co-I(s)) and their staff for establishing and maintaining the 46 sites used in this investigation.


**Financial Statement**

This work is funded by the NASA Modeling, Analysis, and Prediction (MAP) program (PM: David Considine) under the Chemistry-Climate Modeling (CCM) project.




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
