# Peer review of "From column to surface: connecting the performance in simulating aerosol optical properties and PM2.5 concentrations in the NASA GEOSCCM"

_EGUsphere, 2025_

## Author Comment (AC1)

**Response to all referee comments (RCs) - MS No. egusphere-2025-2354**

Many thanks to our referees for taking the time to give us thoughtful comments on our manuscript. Here we provide responses to these referee comments: referee comments are shown in bold, and our responses in plain text. The parts of the manuscript revised in response to comments are in italics here and appear as tracked changes in red in the revised manuscript.

**Response to Referee 1 Comments (RC1)**

**This study combines long-term AOD observations and detailed ground-based PM$_{2.5}$ measurements to comprehensively evaluate GEOSCCM performance in surface aerosol, column AOD, and aerosol optical properties. It identifies key factors for improving performance of GEOSCCM and potentially other models. I recommend this manuscript to be published after minor revisions that address questions below.**

**Specific comments:**

**L159-160: The 'Southern Asia' region is huge and include polluted regions with different behaviors (South Asia vs East Asia). Should justify why they can be combined in this analysis.**

We have added the following clarification in Section 2.2:

*The regional aggregation follows the approach from Collow et al. (2024), which was designed to capture continental-scale trends and behaviors in aerosol distributions. In particular, South Asia and East Asia are grouped together because both are dominated by anthropogenic aerosol sources, particularly from major populated regions in India and China. While we acknowledge that there are important sub-regional differences within this large domain, the continental-scale analysis allows us to examine broad patterns in model performance across regions with similar dominant source characteristics.*

**Section 4.2:**

**Surface PM$_{2.5}$ is notably underestimated for most regions except for NA and Aus, despite that AOD is in good agreement with satellite. Could add some discussions about why surface PM tend to be underestimated by GEOSCCM. Are there any previous work on the aerosol vertical distribution that can explain the bias? What could be done in the future for improving the agreement near the surface?**

The underestimation of surface PM$_{2.5}$ despite reasonable AOD agreement in the non-biomass burning regions likely stems from multiple factors. These could be linked to emission inventory deficiencies, particularly for anthropogenic sources that contribute to systematic biases in several regions. Additionally, the PM2.5–AOD relationship is strongly influenced by aerosol vertical profile (Zhu et al. 2024), and previous comparison with observed aerosol vertical profiles have shown that global atmospheric models can simulate aerosol scale heights higher than observed and transport excessive aerosol mass to the free troposphere at the expense of near-surface concentrations (Koffi et al. 2016; Yu et al. 2010). This vertical redistribution could contribute to explain why column-integrated AOD appears reasonable while surface PM$_{2.5}$ remains underestimated.

Future improvements should focus on updating emission inventories as we outlined in the Conclusions but also carry out a comprehensive evaluation of aerosol vertical profile in the model, which is also fundamental in the mass-optical relationship from the surface to the column integrated AOD.

We have added the following paragraph in section 5:

*The underestimation of surface PM$_{2.5}$ despite reasonable AOD agreement in non-biomass burning regions likely stems from multiple factors. Emission inventory deficiencies, particularly for anthropogenic sources, may contribute to systematic regional biases. Additionally, the PM$_{2.5}$–AOD relationship depends heavily on aerosol vertical distribution (Zhu et al. 2024), and previous studies show that global models tend to simulate excessive aerosol transport to the free troposphere at the expense of near-surface concentrations (Koffi et al. 2016; Yu et al. 2010). This vertical redistribution may contribute to explain why column-integrated AOD appears reasonable while surface PM$_{2.5}$ remains underestimated.*

And the following in the conclusions:

*In addition, further systematic assessment of the model's aerosol vertical profile representation against lidar observations will be important and complementary to this work for understanding the aerosol mass-optical relationship, as well as the connection between surface concentrations and column-integrated AOD retrievals.*

**Section 4.3:**

**(1) the under estimated BC in Figure 6 and good agreement in Figure 7 seem to be suggesting anthropogenic BC not being capture in the model (emission inventory).**

As mentioned in Section 4.3 IMPROVE sites are representative of background concentrations, while the satellite-derived PM$_{2.5}$ are representative of different chemical environments (i.e., background, rural, urban) and so include proximity to stronger anthropogenic sources. We have added the following clarification for BC in section 4.3:

*The model underestimation of BC when compared with satellite derived component but not when compared with the IMPROVE ground-based stations, might indicate that the model is missing some anthropogenic BC in the emission inventory used.*

**(2) There is consistent seasonality bias. And according to IMPROVE, the bias in background dust is getting higher. What could be contributing to this bias and trend? Inland intrusion of dust from the Atlantic Ocean can explain the bias in southeast US and maybe Southern US, but no so much in southwestern US.**

While the seasonal bias in the southeast US can be explained by overestimated intrusion of Saharan dust from the Atlantic Ocean, in the southwest the seasonal positive dust bias could be attributable to an overestimation of local emission sources, such as the Chihuahuan Desert and the Mojave Desert as shown in Figure 7b and Supplemental Figure 13. In addition the model doesn't account for land-use changes that affect dust emissions over time, while observations can capture these evolving conditions. This likely explains the increasing bias between modeled and observed dust concentrations especially in the final years of the study period.

We have modified the following sentence in Section 4.3 from:

*This is most pronounced in the summer, and may be linked to excessive inland intrusion of dust and sea salt mainly from the Atlantic Ocean (Figure 7b and Supplemental Figure 13).*

to:

*This is most pronounced in the summer, and may be linked to excessive inland intrusion of dust and sea salt mainly from the Atlantic Ocean and an overestimation of local dust sources mostly from the Mojave and possibly the Chihuahuan Deserts (Figure 7b and Supplemental Figure 13).In addition the model doesn't account for land-use changes that affect dust emissions over time, while observations capture these evolving conditions. This likely explains the increasing bias between modeled and observed dust concentrations, especially in the final years of the study period.*

**Minor comments:**

**L27: 'relate more to simulated aerosol mass' - would be more accurate to say 'aerosol speciation'**

Corrected as '*simulated aerosol speciated mass'.*

**~L300: eqn 1, sigma simple shows up as a question mark in the pdf document.**

Corrected.

**Table 1: 'Washington U. S. Louis' - it is not common to abbreviate 'St. Louis' as 'S. Louis'.**

Corrected as suggested.

**Figure 6: y axis label is cut off for the first and fourth rows.**

Corrected.

**Table 2: could be moved to the supplements as it is not discussed a lot and figure 8 is illustrative enough.**

Moved Table 2 in the Supplement as Supplemental Table 5.

**L484: 'Dame' should be 'Same'**

Corrected.

**L501: 'except Europe' could be removed since the model doesn't obviously perform worse here.**

Corrected as suggested.

**Response to Referee 2 Comments (RC2)**

**This study presents a comprehensive evaluation of aerosol optical properties (AOD) and PM$_{2.5}$ mass concentration simulations using the GEOSCCM model, combined with multi-source observational data (satellite AOD, ground-based PM$_{2.5}$ measurements, etc.). The work is organized and written in a very clear way. The long-term simulation (1960-2020) and the analysis of seasonal variations are impressive. I recommend the publication of this work after minor revisions.**

**(1) The manuscript compares model outputs with MODIS NNR, AERONET, and IMPROVE datasets**

**and have adequately discussed the reasons of model biases, including emission inventory, and the method simulating inorganic nitrates etc. I suggest that the authors add a few sentences on the uncertainties in observations that might affect the model evaluation, e.g., would MODIS NNR retrievals exhibit biases in biomass burning regions due to differences in smoke optical properties and thus contribute to the model's underestimation of AOD?**

We acknowledge that observational uncertainties may influence our model evaluation results. In general, satellite retrievals of aerosols may be subject to uncertainties due to cloud contamination, inadequate representation of surface reflectance, and aerosol model assumptions (e.g., particle composition and size). The MODIS Neural Network Retrieval (NNR) approach used in our study avoids some of these uncertainties. The NNR translates cloud-cleared observed radiances into AERONET-calibrated AOD without making direct assumptions about aerosol optical properties, which should minimize potential biases in biomass burning regions compared to traditional retrieval methods that rely on predetermined optical property assumptions.

We have added the following text to section 3.1:

*In general, satellite retrievals of aerosols may be subject to uncertainties due to cloud contamination, inadequate representation of surface reflectance, and retrieval aerosol model assumptions (e.g., particle composition and size). The NNR retrieval are less susceptible to these uncertainties than traditional retrievals because it connects the observed radiances to directly observed AOD measurements and so does not explicitly invoke an aerosol model that assumes particle properties.*

Ground-based observations from AERONET and IMPROVE networks introduce uncertainty through representativeness errors, as point measurements are compared against model grid cells spanning 1°×1°. We address this spatial mismatch by emphasizing seasonal and monthly statistical comparisons using long-term observational records.

IMPROVE measurements are additionally susceptible to sampling biases, particularly through volatilization of semi-volatile species during collection and sample handling processes. Semi-volatile compounds such as ammonium nitrate can undergo evaporative losses during filter sampling, resulting in systematic underestimation of nitrate mass and total $PM_{2.5}$ concentrations (Ward et al. 2025; J.L. Hand 2023).

We have added the following text at the end of Section 3.4:

*An important source of uncertainty in our model-observations comparison with the IMPROVE data (as well as the AERONET data) stems from the representativeness error when comparing point measurements with model grid cells (1°×1°). Focusing our comparisons on seasonal and monthly statistics derived from long-term observations can help mitigate this sampling mismatch.*

*Additionally, IMPROVE measurements can be subjected to sampling artifacts, particularly volatilization losses during particle collection and handling. Ammonium nitrate, for instance, can evaporate during filter sampling, leading to underestimation of nitrate concentrations and total $PM_{2.5}$ mass (Ward et al. 2025; J.L. Hand 2023).*

**(2) As RH plays an important role in simulating aerosol scattering, I am curious how to improve RH performance in GEOSCCM?**

Model resolution and explicit representation of land-atmosphere interactions could both improve the representation of RH in models like GEOSCCM. We have added the following text in the conclusions:

*For free-running meteorology models like GEOSCCM, increased spatial resolution could improve convective representation and relative humidity fields, leading to better aerosol extinction simulations (Bian et al. 2009), though computational costs must be balanced against increased accuracy. Additionally, incorporating land-atmosphere coupling could better represent surface processes affecting atmospheric RH through enhanced soil-vegetation-atmosphere feedbacks (Santanello et al. 2018).*

**(3) The current "Discussion and Conclusions" section is long and the readers may miss the main points delivered. I suggest divide section 5 into two sections for discussion and conclusion separately.**

We have separated the current section in section 5 Discussion and section 6 Conclusion.

**(4) Is the simulation presented in Fig. 8(b) and Fig. 10(a) also at RH=35 %? Better clarify this.**

Simulated total $PM_{2.5}$ Fig 8(b) is indeed calculated at RH=35% since it is compared with the IMPROVE gravimetric measurement of fine $PM_{2.5}$. Individual components are measured dry through Ion Chromatography, Thermal Optical Reflectance or X-Ray Fluorescence depending on the species (J.L. Hand 2023) so we compared simulated dry mass for the speciated $PM_{2.5}$ in Figure 10(a) . We have added clarification in each of these subplots if measurement and simulation of mass is at RH=35% or dry (RH=0%).

**Bibliography**

Bian, H., M. Chin, J. M. Rodriguez, H. Yu, J. E. Penner, and S. Strahan. 2009. "Sensitivity of Aerosol Optical Thickness and Aerosol Direct Radiative Effect to Relative Humidity." *Atmospheric Chemistry and Physics* 9 (7): 2375–86. https://doi.org/10.5194/acp-9-2375-2009.

J.L. Hand. 2023. "IMPROVE DATA USER GUIDE 2023 (VERSION 2)." https://vista.cira.colostate.edu/Improve/data-user-guide/.

Koffi, Brigitte, Michael Schulz, François-Marie Bréon, et al. 2016. "Evaluation of the Aerosol Vertical Distribution in Global Aerosol Models through Comparison against CALIOP Measurements: AeroCom Phase II Results." *Journal of Geophysical Research: Atmospheres* 121 (12): 7254–83. https://doi.org/10.1002/2015JD024639.

Santanello, Joseph A., Paul A. Dirmeyer, Craig R. Ferguson, et al. 2018. *Land–Atmosphere Interactions: The LoCo Perspective*. Bulletin of the American Meteorological Society. June 1. https://doi.org/10.1175/BAMS-D-17-0001.1.

Ward, Ryan X., Haroula D. Baliaka, Benjamin C. Schulze, et al. 2025. "Poorly Quantified Trends in Ammonium Nitrate Remain Critical to Understand Future Urban Aerosol Control Strategies." *Science Advances* 11 (21): eadt8957. https://doi.org/10.1126/sciadv.adt8957.

Yu, Hongbin, Mian Chin, David M. Winker, et al. 2010. "Global View of Aerosol Vertical Distributions from CALIPSO Lidar Measurements and GOCART Simulations: Regional and Seasonal Variations." *Journal of Geophysical Research: Atmospheres* 115 (D4). https://doi.org/10.1029/2009JD013364.

Zhu, Haihui, Randall V. Martin, Aaron van Donkelaar, et al. 2024. "Importance of Aerosol Composition and Aerosol Vertical Profiles in Global Spatial Variation in the Relationship between $PM_{2.5}$ and Aerosol Optical Depth." *Atmospheric Chemistry and Physics* 24 (20): 11565–84. https://doi.org/10.5194/acp-24-11565-2024.